# Differences in 6-Minute Walk Distance Across Heart Disease Recurrence Risk Levels in Cardiac Rehab Patients

**DOI:** 10.3390/healthcare12222280

**Published:** 2024-11-15

**Authors:** Eric Lian, Kimberly Roberts, Lufei Young

**Affiliations:** 1Medical College of Georgia, Augusta University, Augusta, GA 30912, USA; 2Wellstar Health System, Marietta, GA 30066, USA; 3School of Nursing, University of North Carolina, Charlotte, NC 28223, USA

**Keywords:** cardiac rehabilitation, six-minute walk distance, 2-year recurrent heart disease, cardiac risk

## Abstract

Background/Objectives: Cardiac rehabilitation (CR) programs are multi-component interventions comprising structured strength and cardiovascular exercise training, psychological support, education, and therapies to promote positive lifestyle changes. This study aimed to determine if there are differences in the 6-minute walk distance (6MWD) across risk groups for recurrent heart disease. Methods: This retrospective cohort study used existing data collected from electronic medical records. The 6-minute walk distance was measured at baseline (pre-6MWD) and upon the completion of the CR program (post-6MWD). Short-term cardiac event recurrence risk was determined using a two-year recurrent coronary heart disease (2yRCHD) risk percentage, calculated according to the Framingham 2yRCHD calculator. Risk was then stratified into (1) low, (2) moderate, and (3) high-risk groups. Demographic variables (e.g., age, sex, racial/ethnic group) and clinical variables (e.g., BMI, lipid panels, fasting glucose levels, comorbidities) were collected to describe the study participants and identify potential confounders. An ANOVA and ANCOVA were performed to examine the differences in 6MWD across the 2yRCHD risk groups. Results: A total of 394 CR participants’ data were included in this analysis. Ninety-nine percent of the female participants were classified as low risk for recurrent heart disease, resulting in an extremely small sample size in the moderate-risk (n = 1) and no representation (n = 0) in the high-risk group. This lack of representation made it impossible to conduct comparative analyses across all the participants or to analyze female participants separately by risk category. Consequently, only male participant data were included in the final analysis. The study showed that pre- and post-6MWD measurements were significantly different across the three 2yRCHD risk groups (*p* = 0.006 for pre-6MWD; *p* = 0.002 for post-6MWD). The ANCOVA indicated that these differences were independent of the selected covariates. Post hoc analyses revealed significant differences in 6MWDs between the low- and high-risk groups and between the moderate- and high-risk groups, but not between the low- and moderate-risk groups, for both pre- and post-6MWD measurements. Compared to the CR participants in the high-risk group, those in the low- and moderate-risk groups achieved significantly longer distances in the 6-minute walk tests. Conclusions: The observed differences in the 6MWD across short-term cardiac recurrence risk levels suggest its potential as a simple, accessible tool for assessing cardiac recurrence risk levels in community settings. Further research is needed to generalize these findings to more diverse populations and to support aging in place for older adults living with heart disease.

## 1. Introduction

Cardiac rehabilitation (CR) programs are multi-component interventions that comprise structured strength and cardiovascular exercise training, psychological support, education, and therapies to promote positive lifestyle changes [1]. Its goal is to reduce cardiovascular risk factors, mortality, morbidity, and improve the quality of life in cardiac patients [2]. It is essential post-operative care for individuals recovering from cardiac surgeries and procedures, as well as an important aspect of care following cardiac events. Each patient’s data during cardiac rehabilitation is documented to track and assess their progress throughout the program [3]. CR program consists of three phases: inpatient care (phase 1) focused on early recovery and education, an outpatient program (phase 2) with supervised exercise and lifestyle training, and a long-term maintenance phase (phase 3) to help patients sustain heart-healthy habits independently [4]. It was reported that CR participation was associated with a 32% risk reduction in all-cause mortality, indicating the crucial role of CR participation in secondary prevention among patients with various cardiovascular diseases [5]. This eventually led to Medicare approval for CR reimbursement. CR programs are the centers of lifestyle modifications, which have improved the outcomes of post-cardiac event patients [6].

The evaluation of the effects of a cardiac rehabilitation (CR) program on cardiovascular health requires accurate assessment tools. The gold standard for measuring direct cardiovascular fitness is maximal oxygen consumption (VO_2_ max) [7]. VO_2_ max provides a comprehensive assessment of all of the systems during the exercise (i.e., the heart, circulation, lungs, and musculoskeletal system) and is a reliable indicator of risk for recurrent cardiac events [8]. Despite VO_2_ max being a preferred assessment for cardiac risk, it is challenging to administer in older adults participating in CR programs [9,10]. VO_2_ max testing requires individuals to exercise at progressively higher intensities until they reach maximal exertion, which can be too physically demanding for CR participants with preexisting cardiac conditions, increasing the risk of adverse events [9,10]. In many cases, safety concerns outweigh the benefits of obtaining a precise VO_2_ max reading. Additionally, the CR participants may have had other physical limitations or chronic conditions that impair their ability to perform the intense exercise required for an accurate VO_2_ max test [9,10]. VO_2_ max testing also requires specialized equipment and trained personnel to monitor the test and ensure safety, adding logistical and financial barriers that may be difficult to justify in many cardiac rehabilitation settings [9]. Due to these challenges, submaximal exercise tests, such as the 6-minute walk distance (6MWD) test, are often used in CR programs.

CR programs use the 6-minute walk distance (6MWD) test to evaluate participants’ functional capacity, exercise tolerance, and endurance. This test requires participants to walk as far as possible in six minutes on a flat, straight surface, with the total distance covered recorded as the result [11,12]. The 6MWD test is repeated periodically throughout the CR program to tailor exercise prescriptions, set achievable goals, and track progress [12]. At the end of the program, an improved 6MWD score may indicate that the participant is ready to transition out of supervised CR and continue with a self-managed exercise plan [12]. A substantial body of literature supports the correlation between 6MWD and VO_2_ max, with the 6MWD test often used to estimate VO_2_ max [10,13,14]. While VO_2_ max is the gold standard for assessing cardiac recurrence risk, it is not always feasible for CR participants. Consequently, 6MWD may serve as an alternative measure for assessing recurrent cardiac event risk, though no evidence has yet been reported specifically in CR participants. To fill this gap in knowledge, the objective of this analysis was to determine whether the 6-minute walk distance differs across various risk groups for recurrent heart disease using the data collected from CR participants.

## 2. Materials and Methods

### 2.1. Study Design

This was a retrospective, observational cohort study using existing data collected from the electronic medical record (EMR) at a cardiac rehabilitation center affiliated with a large medical center in the southeastern United States. This study was approved by the Institutional Review Board (IRB). This manuscript was drafted in accordance with the STROBE (Strengthening the Reporting of Observational Studies in Epidemiology) guidelines to ensure comprehensive and transparent reporting [15,16].

### 2.2. Setting

This study was performed at a large not-for-profit academic medical center located in the southeastern United States. The outpatient cardiac rehabilitation center was part of this medical center and provided rehabilitation care to the patients with cardiovascular disease starting at the age of 17 years. The CR program addressed the physical, psychosocial, and educational needs of the patients with heart disease. With a focus on lifestyle modification, the CR program assisted patients as they recovered from myocardial infarction, other forms of cardiac condition, or surgery. The patients were referred by their physician and received a medical evaluation, physical activity training, and lifestyle education, and support. The existing data collected from the patients who attended this cardiac rehabilitation center were used for this analysis.

### 2.3. Participants and Data Sources

This study’s participants (n = 394) were patients who attended the CR program. Their data were collected from the Individual Cardiac Treatment Plan (ICTP) records. An ICTP is a personalized care plan developed for CR participants, including medical history, risk factors, cardiac condition, baseline and ongoing physical assessment (i.e., 6MWD tests, limb strength), and clinical assessment (i.e., vital signs, body mass index [BMI]), as well as exercise prescriptions, nutritional guidance, medication list, mental health counseling, and support. The CR patients’ data were included if they completed all 36 CR sessions and graduated from the program with both pre- and post-6MWD test results in ICTP form. The patients who had incomplete or missing ICTP records and failed to complete the CR program were excluded from this study.

### 2.4. Bias and Sample Size

To minimize selection bias, we included all the available CR participants from the cardiac rehabilitation center who had complete pre- and post-6MWD test results recorded in the electronic medical records (EMR) to obtain a representative sample. Information bias was addressed by applying strict inclusion criteria for data completeness. We also verified that the 6MWD test was conducted according to standard procedures within the CR program to ensure consistency in measurements. Potential confounding factors (e.g., age, sex, comorbidities) were collected and adjusted for in the analysis using an ANCOVA. The sample size was determined based on the availability of the retrospective EMR data from the patients who completed the CR program with recorded pre- and post-6MWD test results. Since this was a retrospective analysis, we did not conduct a formal sample size calculation prior to data collection. Instead, we utilized all the eligible cases within the specified time frame to maximize the statistical power for detecting differences in the 6MWDs across the low-, moderate-, and high-risk groups for recurrent cardiac events.

### 2.5. Variables and Measures

The baseline 6MWD (pre-6MWD) and the last 6MWD at the completion of the CR program (post-6MWD) were collected. The risk of a recurrent cardiac event was assessed using the 2-year recurrent heart disease risk (2yRCHD risk) score [12]. The Framingham Risk Score (FRS) is a widely used tool developed based on data from the Framingham Heart Study to estimate the risk of developing cardiovascular disease (CVD) [17]. It has been adapted to assess shorter-term risks, including the 2-year recurrent heart disease risk (2yRCHD) in individuals who have already experienced a cardiac event [12]. The clinical factors used to calculate 2yRCHD risk include age, gender, hypertension, cholesterol levels, smoking status, diabetes, previous cardiac events and the use of medications. The 2-year risk score can be calculated using a point system based on the above risk factors. Each factor is assigned a certain number of points depending on its severity or presence. For instance, points increase with age (e.g., 2 points for age 55–64, 3 points for 65–74, etc.). Higher cholesterol level, systolic blood pressure, current smoking, and having diabetes contribute more points. Once the points for each risk factor were totaled for each participant, the scores were assigned into different risk groups (low, moderate, high) based on the predetermined parameter [17]. 2yRCHD helps identify individuals who are at increased risk of another cardiac event within two years.

### 2.6. Statistical Analyses

All the statistical analyses were calculated using IBM SPSS Statistics (Version 27) [18] and the statistical significance was assessed using an alpha level of 0.05. Descriptive statistics (means, standard deviations, percentages, and frequencies) were calculated for all the variables. The independent variable of interest in this study was a 2yRCHD risk, which included three groups: low, moderate, and high. Low was defined as having a <10% chance of recurrent heart disease within 2 years according to the Framingham Recurrent Coronary Heart Disease Risk Calculator. Moderate was defined as having a 10–15% 2yRCHD risk, and high was defined as having a >15% 2yRCHD risk. The dependent variables of interest were pre-6MWD and post-6MWD.

To analyze the differences in 6MWD across the low-, moderate-, and high-risk groups for 2-year recurrent coronary heart disease (2yRCHD), we applied both an Analysis of Variance (ANOVA) and an Analysis of Covariance (ANCOVA). The ANOVA was used to assess the overall differences in 6MWD across the risk groups. To control for potential confounding factors (e.g., age, sex, BMI, and comorbidities), an ANCOVA was conducted to adjust the 6MWD results based on these covariates, allowing us to isolate the effect of the risk group on the 6MWD. We also examined the interaction effects between the risk groups and the covariates in the ANCOVA to ensure that the observed relationships were consistent across various demographic and clinical variables.

To address missing data, we applied strict inclusion criteria, only including cases with complete pre- and post-6MWD measurements and relevant covariates in the analysis. Cases with missing data for key variables were excluded to maintain consistency and accuracy in the final results. This study utilized retrospective data from the electronic medical records of the CR participants, which were de-identified; therefore, there was no active follow-up phase or risk of loss to follow-up. Only the participants who completed the CR program and had both pre- and post-6MWD data were included in the analysis.

Sensitivity analyses were conducted by comparing the results of the ANOVA and ANCOVA to assess the robustness of the findings with and without adjustment for covariates. Additionally, the results were compared across different demographic groups to confirm consistency in the observed differences in 6MWD across the risk groups. We also compared the CR participants with complete pre- and post-6MWD data to those with missing data who were excluded from this analysis. There were no significant differences in all the available variables between the included CR participants with complete pre- and post-6MWD data and the excluded participants with missing data. This enhances our confidence that our findings have minimal selection bias due to missing data.

## 3. Results

### 3.1. Participant Characteristics

The participant characteristics are presented in Table 1. A total of 394 CR participants’ data were included in this analysis. The mean age of the sample was 62 years (SD = 12.15; range 18–91 years). The participants included 263 (66.8%) males and 131 (33.2%) females. The racial demographics of the participants included 231 (59%) white people, 132 (34%) Black people, and 31 (3%) people of other races. The average BMI at baseline was 30.43 (SD = 6.72), indicating that most of the participants were overweight or obese. The female participants had a higher average BMI than the male counterparts. The average ejection fraction at baseline was 50% (SD = 14.21%). The participants had an average of five chronic conditions (SD = 2.98). The types of chronic conditions most prevalent in the population were hypertension (n = 195, 49.52%), hypercholesterolemia (n = 109, 27.95%), diabetes (n = 67, 17.14%), obstructive sleep apnea (n = 37, 9.44%), and cerebral vascular accident (n = 27, 6.85%). A history of obstructive sleep apnea (OSA) was statistically significant for the males (χ^2^ = 5.38, *p* = 0.02) as compared to the females. About 36% (132) of the CR participants were tobacco users at baseline when they were enrolled in the CR program. There was a statistically significant difference in tobacco use between males and females (χ^2^ = 1.71, *p* = 0.042), with a higher prevalence of tobacco use among males.

### 3.2. Pre- and Post-Cardiac Rehabilitation Comparisons

Among all the CR participants included in this analysis, the average 6MWD significantly increased from 1321 feet at baseline to 1525 feet after CR completion (t = 16.582; *p* < 0.001). In the male participants, the average 6MWD significantly increased from 1365 to 1558 feet (t = 13.732; *p* < 0.001). In the female participants, the average 6MWD significantly increased from 1234 to 1443 feet (t = 9.379; *p* < 0.001). A significant sex difference was observed in both the pre- and post-6MWDs. The female participants had lower performance improvement in the 6MWD tests than the male participants (Table 2).

Besides the primary variable, 6MWD, we also compared the changes in several cardiac risk factors before and after the completion of the CR program (Table 2). The average weight increased significantly among all participants (t = 2.635, *p* = 0.004) and specifically among male CR participants (t = 2.586, *p* = 0.005), but not among female CR participants (t = 0.752, *p* = 0.227). There was no significant improvement in body fat percentage among all the participants (t = 1.058, *p* = 0.146), the male participants (t = 1.111, *p* = 0.134), or the female participants (t = 0.545, *p* = 0.294). Similarly, there was no significant improvement in the total cholesterol levels among all the participants (t = -0.434, *p* = 0.332), the male participants (t = -0.591, *p* = 0.278), or the female participants (t = 0.630, *p* = 0.265). The fasting glucose levels significantly decreased in all the participants (t = 2.037, *p* = 0.022) and in the female participants (t = 2.158, *p* = 0.018), but not in the male participants (t = 1.499, *p* = 0.068). There was no significant improvement in the Hemoglobin A1C levels among all the participants (t = 1.068, *p* = 0.143), the male participants (t = 0.651, *p* = 0.258), or the female participants (t = 0.847, *p* = 0.2). In summary, our findings did not demonstrate a statistically significant reduction in cardiac risk factors (i.e., body fat percentage, total cholesterol level, fasting blood glucose, and Hemoglobin A1C) following the completion of the CR program in this cohort.

### 3.3. Pre- and Post-6MWDs Across 2yRCHD Risk Groups

Table 3 presents the distribution of CR participants and their pre- and post-6MWD performance based on their 2-year recurrent heart disease (2yRCHD) risk groups (low, moderate, high). Among the 394 CR participants, 59% (234) were classified as low-risk, 33% (131) as moderate-risk, and 7% (29) as high-risk. Nearly half of the male participants (50%, 130) were in the moderate-risk group, with 40% (104) in the low-risk group and 11% (29) in the high-risk group (Figure 1). In contrast, almost all the female participants (99%, 130) were classified as low-risk, with only one female in the moderate-risk group and none in the high-risk group. The participants in the low- and moderate-risk groups had higher pre- and post-6MWDs compared to those in the high-risk group.

### 3.4. Pre- and Post-6MWDs Across 2yRCHD Risk Groups in Male

Ninety-nine percent of the female participants (n = 130) were classified as low-risk for the 2-year risk of recurrent coronary heart disease (2yRCHD), with only one female in the moderate-risk group and none in the high-risk group. As a result, group comparisons among the female participants were not feasible due to the extremely small sample size in the moderate-risk group and the absence of participants in the high-risk group. Consequently, it was not appropriate to include females in the analysis to determine the differences in the 6MWD across the three risk groups for 2yRCHD. Therefore, the analysis was conducted using data from the male participants only.

Table 4 presents the descriptive statistics for the pre- and post-6MWDs across each 2yRCHD risk group in the male participants, as well as the results of the One-Way ANOVA. There was a statistically significant difference in the pre-6MWDs among the three 2yRCHD risk groups (low, moderate, high), with shorter 6MWDs in the higher risk group (F = 6.312, *p* = 0.006). Similarly, a statistically significant difference was observed in the post-6MWDs among the three risk groups, with shorter 6MWDs in the higher risk group (F = 4.882, *p* = 0.027). After adjusting for multiple comparisons, post hoc analyses revealed significant differences in the 6MWDs between the low- and high-risk groups and between the moderate- and high-risk groups, but not between the low- and moderate-risk groups, for both the pre- and post-6MWD measurements. Compared to the CR participants in the high-risk group, those in the low- and moderate-risk groups achieved significantly longer distances in the 6 min walk tests.

There were no significant differences in the results between the ANOVA and ANCOVA. This indicates that the covariates included in the ANCOVA (e.g., age, BMI, comorbidities, and the other clinical variables listed in Table 1 and Table 2) did not substantially impact the outcome, suggesting that they did not contribute additional explanatory power beyond what was captured by the ANOVA alone.

## 4. Discussion

This analysis was conducted using the existing data from the patients who attended a single-center cardiac rehabilitation (CR) program. The objective of this analysis was to determine whether the 6 minute-walk distance (6MWD) significantly differs across low-, moderate-, and high-risk groups for short-term recurrent cardiac events. We used the Framingham 2yRCHD calculator to determine and assign risk group membership to the study participants. Multi-group comparisons were conducted using the ANOVA and ANCOVA for both the baseline 6MWDs (pre-) and 6MWDs (post-) after 3 months of CR program. The results indicated that both the pre- and post-6MWDs were significantly different across the low-, moderate-, and high-risk groups. The results demonstrate that the male CR participants in the low- and moderate-risk groups had significantly greater pre- and post-6MWD compared to those in the high-risk group.

The 6MWD test demonstrated a significant difference across the low-, moderate-, and high-risk groups, highlighting its utility as a simple, accessible tool for distinguishing levels of cardiac risk in community settings. Our findings align with previous research that supports the 6MWD as a valuable indicator of cardiovascular functioning, further underscoring the role of the 6MWD in differentiating short-term recurrent cardiac event risk levels among CR participants. This study contributes novel evidence suggesting that 6MWD may be capable of distinguishing between varying levels of recurrent cardiac risk in CR participants, adding to the body of research advocating for functional capacity tests in assessing long-term cardiac prognosis beyond the rehabilitation period. By incorporating 6MWD as a routine assessment tool, healthcare providers can more effectively monitor and tailor interventions to each patient’s risk level.

The findings of this study have important implications for clinical practice and research. While our results show the potential use of the 6MWD test as a risk assessment tool for short-term cardiac event recurrence, it is best used as a supplementary measure alongside traditional risk assessment tools rather than as a replacement.

This study has several limitations. First, it was a secondary data analysis using electronic medical record (EMR) data which usually contains missing data. We employed multiple strategies to minimize the potential impact of missing data. First, we applied strict inclusion criteria, only including cases with complete pre- and post-6MWD measurements and relevant covariates in the analysis. We also compared the CR participants with complete pre- and post-6MWD data to those with missing data who were excluded from this analysis, finding no significant differences between the two groups. Additionally, we compared the results across different demographic groups to confirm consistency in the observed differences in 6MWD across the risk groups. Sensitivity analyses were conducted by comparing the results of the ANOVA and ANCOVA to assess the findings with and without adjustment for covariates.

Ninety-nine percent of the female participants were classified as low risk for recurrent heart disease, resulting in an extremely small sample size in the moderate-risk group (n = 1) and no representation (n = 0) in the high-risk group. This lack of representation made it impossible to conduct comparative analyses across all the participants or to analyze the female participants separately by risk category. Consequently, only the male participant data were included in the final analysis. Sensitivity analyses indicated minimal selection bias in our data. The absence of female participants in the high- and moderate-risk groups likely may be due to sex differences in the pathophysiology and prognosis of the cardiovascular diseases. It may reflect underlying mechanisms in which males and females exhibit distinct patterns of disease development, recovery, and rehabilitation. These sex-based differences are rarely explored from a biological perspective in current CR studies, which often focus instead on the differences in the knowledge acquisition and behavioral changes between men and women. As a result, the impact of sex on CR program outcomes remains underreported, leaving a gap in understanding how male and female participants may respond differently to cardiac rehabilitation. Another consideration is the measurement itself. The 2-year recurrent heart disease (2yRCHD) calculation, derived from the Framingham study, was developed based on commonly accepted cardiac risk factors, with variable weighting across the included factors. It is possible that females have unique risk factor profiles or require alternative weighting for cardiac risk, which remains an area of limited understanding. Including female data with no representation in the high-risk group prompts readers to consider the potential moderating effect of sex on cardiovascular disease prognosis and responses to CR programs. Future research specifically targeting female participants is needed to better understand the sex-based differences in cardiac risk assessment.

We used de-identified data, making it impossible to re-link the individual participants to track their actual cardiac events. The findings of this analysis highlight the potential of the 6MWD as a risk assessment tool for short-term recurrent cardiac events. In future studies, we aim to enhance validity by conducting a prospective, longitudinal study with a control group to address the limitations of this retrospective secondary data analysis. Alongside the repeated measures of the 6MWD, we will include assessments of frailty, inflammation, nutritional markers, and VO2 max—the gold standard for cardiovascular fitness. This triangulation approach will provide a more comprehensive and reliable evaluation of CR’s long-term impact. Such data would enable a deeper understanding of patient prognosis and the progression of risk category changes over time, including ongoing assessments of the 6MWD, frailty indices, and VO2 max, along with tracking actual cardiac events, which would help evaluate the reliability and validity of the 6MWD as a potential risk assessment tool. Additionally, exploring the effectiveness of the community-based CR programs using the 6MWD as a monitoring tool, alongside VO2 max, could pave the way for more equitable and sustainable cardiac care, supporting aging in place.

The data were collected from the CR participants at a single center. The patients that were eligible for CR typically had stable conditions and the physical capacity required to engage in structured exercise. The small sample size in the high-risk group likely reflects this selection bias, as the individuals with more severe or unstable conditions were likely excluded. This limitation restricts the generalizability of our findings to populations with more complex conditions and multiple comorbidities. For future research, studies should aim to include a more diverse and larger sample, particularly in high-risk groups and among female participants, to further validate these findings.

## 5. Conclusions

Cardiopulmonary rehabilitation programs are designed to induce structured exercise, lifestyle modifications, and risk reduction strategies in patients following an episode of cardiovascular event [15,16,17]. This retrospective cohort study was conducted to determine if there are differences in the 6 minute-walk distance (6MWD) across the risk groups for recurrent heart disease. Our study offers new insights into the use of the 6MWD as a simple, accessible tool for identifying CR patients with different levels of risk for short-term cardiac event recurrence. Further research is needed to generalize these findings to more diverse populations and to optimize care for all heart disease patients, particularly those aging in place. Future studies should include a more diverse and larger sample, particularly in high-risk groups and among female participants, to further validate these findings.

## Figures and Tables

**Figure 1 healthcare-12-02280-f001:**
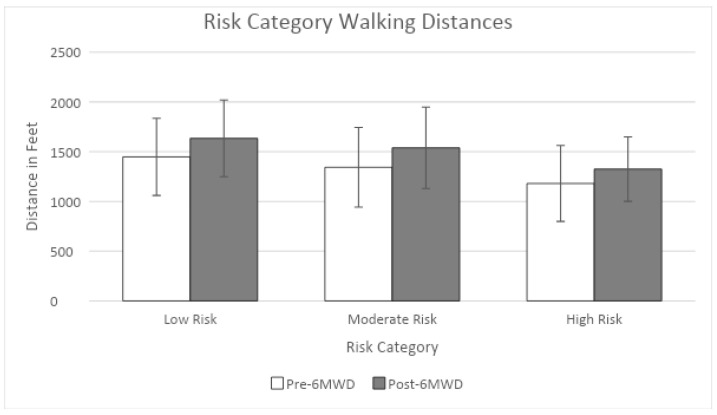
Pre- and post-6MWD across risk categories in males.

**Table 1 healthcare-12-02280-t001:** Participant characteristics (N = 394).

Variable Names	Overall M ± SD or N (%)	Sex M ± SD or N (%)	
		**Male**	**Female**
Number	394	263 (66.8)	131 (33.2)
Age	62.44 ± 12.15	63.06 ± 11.77	61.18 ± 12.85
Race			
Black	132(33.67)	71(54.20)	54(41.22)
White	231(58.93)	160(61.07)	71(54.20)
BMI in kg/m^2^	30.43 ± 6.72	30.28 ± 6.44	30.74 ± 7.27
Ejection fraction (%)	50.07 ± 14.21	49.47 ± 14.24	51.31 ± 14.14
Hypertension (yes)	195 (49.62)	137 (52.09)	58 (44.27)
Hyperlipidemia (yes)	109 (27.95)	74 (28.46)	35 (26.72)
Diabetes (yes)	67 (17.14)	45 (17.56)	22 (16.15)
OSA (yes)	37 (9.44)	31 (11.83) *	6 (4.58)
CVA (yes)	27 (6.85)	18 (6.84)	9 (6.87)
Tobacco use (yes)	132 (36)	94 (38.68) *	38 (31.71)
No. chronic conditions	5.38 ± 2.98	5.5 ± 3.04	5.11 ± 2.85

**Note**: M, means; SD, standard deviation; BMI, body mass index; OSA, obstructive sleep apnea; CVA, cerebrovascular accident. * *p* < 0.05

**Table 2 healthcare-12-02280-t002:** Comparison of the cardiac risk factors at baseline and after the completion of cardiac rehabilitation by gender (N = 394).

Variable Names		Overall M ± SD			Male M ± SD			Female M ± SD	
	Baseline	3-Month	*p* Value	Baseline	3-Month	*p* Value	Baseline	3-Month	*p* Value
Weight (lb.)	200.23 ± 52.43	200.82 ± 52.25	*p* = 0.004	210.57 ± 52.9	211.24 ± 53.26	*p* = 0.005	179.54 ± 44.99	179.48 ± 42.99	*p* = 0.227
Body fat percentage	34.24 ± 6.93	34 ± 9.04	*p* = 0.146	31.65 ± 6.04	31.24 ± 7.65	*p* = 0.134	39.42 ± 5.57	39.92 ± 8.98	*p* = 0.294
Total Cholesterol level (mg/dL)	166.94 ± 45.69	164.89 ± 71.42	*p* = 0.332	163.8 ± 44.31	160.8 ± 78.66	*p* = 0.278	172.58 ± 47.79	173.51 ± 52.31	*p* = 0.265
Fasting Glucose level (mg/dL)	121.24 ± 45.86	119.53 ± 48.86	*p* = 0.022	122.23 ± 44.95	119.58 ± 48.58	*p* = 0.068	119.24 ± 47.93	119.45 ± 49.7	*p* = 0.018
Hemoglobin A1C (%)	6.97 ± 6.51	6.89 ± 5.96	*p* = 0.143	7.23 ± 8.06	7.13 ± 7.32	*p* = 0.258	6.52 ± 1.74	6.46 ± 1.46	*p* = 0.2
6MWD (ft.)	1320.95 ± 389.14	1524.68 ± 383.04	*p* < 0.001	1364.96 ± 400.31	1558.29 ± 401.32	*p* < 0.001	1233.63 ± 351.43	1442.65 ± 321.96	*p* < 0.001

Note: M, means; SD, standard deviation; 6MWD, six-minute walk distance in feet.

**Table 3 healthcare-12-02280-t003:** Pre- and post-6MWD distribution across risk groups by gender.

2yRCHD Risk Group		Low	Moderate	High
N (%)	Overall	234(59.4)	131 (33.2)	29 (7.4)
	Male	104 (39.5)	130 (49.4)	29 (11.0)
	Female	130 (99.2)	1 (0.8)	0 (0)
Pre-6MWD	Overall	1327.53 (±382.46)	1341.03 (±399.74)	1180.36 (±381.29)
	Male	1442.72 (±388.10)	1342.78 (±400.94)	1180.36 (±381.29)
	Female	1234.46 (±352.72)	-	-
Post-6MWD	Overall	1546.96 (±368.86)	1535.51 (±409)	1324.88 (±324.50)
	Male	1634.28 (±385.63)	1539.01 (±409.69)	1324.88 (±324.50)
	Female	1445.58 (±322.79)	-	-

**Table 4 healthcare-12-02280-t004:** Differences in pre- and post-6MWD across 3 risk groups in male.

2yRCHD Risk	N	Pre-6MWD Mean (SD)	*p*-Value	Post-6MWD Mean (SD)	*p*-Value
Low	104	1447.72 (388.10)	0.006	1634.28 (385.63)	0.002
Moderate	130	1342.78 (400.94)		1539.01 (409.69)	
High	29	1180.36 (381.29)		1324.88 (324.50)	

Note: 2yRCHD, two-year recurrent heart disease; 6MWD, 6-minute walk distance (feet).

## Data Availability

The data are unavailable due to privacy and ethical restrictions. Additionally, public data sharing was not included in the original IRB protocol. This prohibits our ability to share the data while maintaining compliance with ethical standards.

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
