# Peer review of "Differences in 6-Minute Walk Distance Across Heart Disease Recurrence Risk Levels in Cardiac Rehab Patients"

_healthcare, 2024, doi:10.3390/healthcare12222280_

Round 1

Reviewer 1 Report

Comments and Suggestions for Authors

I read with interest the manuscript 'Association Between Six-Minute Walk Distance and 2-Year Re-2 current heart disease in Male Cardiac Rehab Patients'. 

There are several areas of improvement that can strengthen the study and scientific soundness of the aims. I also include here several technical aspect of better constructing the paper.

Table 1 - to provide full terms of the abbreviations used, and to provide p values to indicate significance of changes/improvement statistically, wherever applicable, between pre- and post- measured values.

Table 2 - to re-format the Table to show whether 6MWD changes significantly pre- and post-CRP; to provide full terms of the abbreviations used

Figures 1 and 2 - to re-format the bar charts; consider recategorizing according to risk groups but aim to show change graphically, and include p values

Figure 3 - to include p values in the diagram

Line 244 and 254 - this has to be substantiated with an area under the curve (AUC) analysis, and the authors to propose a cut-off value for predicting high categories (moderate, high, or moderate + high) 2-year event rate

Line 263 - major contradiction; the authors to include univariate and multivariate analysis to show statistical correction of the 6MWD values, either pre-, post-, or change.

Line 278-287 - this paragraph is subject to the local or regional healthcare models; in most countries, cardiac rehabilitation programs are subsidized either in part or entirely. 6MWD data at best, is additive but cannot supplant traditional risk assessment tools to estimate risk.

Author Response

We have enclosed a revised manuscript with revised title, “Differences in 6-Minute Walk Distance Across Heart Disease Recurrence Risk Levels in Male Cardiac Rehab Patients". 

 In this revised manuscript, we addressed all your reviewers’ comments as described in detail under the heading, Response to Reviewer Comments, on the following pages. We provided an itemized, point-by-point response to the comments of the reviewers. We deeply appreciate your time and effort spent on helping improve our manuscript. All the helpful comments greatly strengthened the manuscript. We used blue ink for all edits and changes to make them easy for the reviewers to find. 

Reviewer 2 Report

Comments and Suggestions for Authors

I would like to congratulate the authors on the relevant paper and research. The Discussion and Conclusion sections are very informative. However, for the Introduction, Methods, and Results, I have a few major comments to make:

Major comments:

  • In the final paragraph of the "Introduction", the authors establish the reason why they conducted the research. However, I’m not sure that I’m convinced by what is written there. The authors state that it is unclear which component of the cardiac rehabilitation (CR) program is responsible for the cardiovascular risk reduction observed in previous meta-analyses. But, as far as I know, the 6-minute walk test is not an intervention component performed in the CR program but rather, literally, a follow-up test of what is being done in the program. To clarify my concern, after reading the article, do the authors believe that the scientific community will now know which component of the CR programs is responsible for the cardiovascular risk reduction? If so, please explain this paragraph more clearly; or, if not, I suggest modifying the rationale for the research in that paragraph.

  • Figures 1 and 2 have a design and graphical quality well below the standards for a scientific publication. Furthermore, they are redundant with the content in Table 2, which is much more informative. I strongly recommend that the authors remove Figures 1 and 2 from the manuscript.

  • Authors should explain in detail how missing data was addressed in the "Methods" section. I strongly recommend following the STROBE guidelines when writing the article.

  • Regarding the missing data on women in your sample, I recommend reading this article: https://doi.org/10.1186/1471-2288-12-96 and discussing with your team of statisticians whether there is a better way to report the data from your research.

A curiosity: Is it possible to conduct a post-hoc analysis of how many patients actually experienced events in your sample? Given that we are dealing with data from patients from the past, I imagine a post-hoc analysis of this could be informative, as long as it is well addressed and the limitations of such an analysis are clearly described in the text.

Author Response

(The authors gave the same response as above.)

Reviewer 3 Report

Comments and Suggestions for Authors

Thank you for the opportunity to review this paper.

Study purpose is to determine any association with improvements in 6min walk time and short term cardiac recurrent risk.

General – The term association implies that you did a correlation analysis, yet I see only ANOVA comparisons. I do not see any post hoc tests after the main effects. I believe you need to examine the statistical analysis and align it with the purpose of the study. It seems that you are examining improvements in cardiac risk variables across the three risk categories following 36 sessions of CR. I do not see any correlation study between 6mw and risk profile.

Introduction- while the introduction does refer to all cause mortality improvement and CR, the study is in particular about the 6min walk test, a marker for VO2max improvement. Can you provide some background in the introduction regarding the importance of VO2max improvements as a key aspect of CR? This relates more to your topic, unless you wish to discuss the components of CR that also relate to all cause mortality. Currently the introduction does not identify any potential variables of interest.

Line 53,77- “heart attacks” is really a lay person term is it not? Perhaps myocardial infarction?

2.2- You should probably mention the total number of subjects in the data pool in this section. I had to go looking for this information, so I think it might  go well under the participants section

Statistical analysis- the phrase “determine the association of age….” By association, I read correlation, but the analysis seems to be looking at improvements in cardiac risk profile following CR.  You results section seems to show pre post changes (ANOVA results) so I’m trying to understand the “association” you are seeking to assess.

Is there anyway to know if any of the participants moved from high risk to moderate or low risk following CR? How did you track where the same subjects ended up post training? If they did improve and move categories, then only the non-responding patents would be left in the high category, and therefore show no change, while others did.

Table 1- this is showing overall data, I can’t see across the risk categories. Can we see the pre-post data across each risk category?

Table 3- Is the P value the main effects of the anova across the categories? This is not really “association” this is comparison.  Figure 3 seems to be a version of table 1. Is it needed? Was an ANOVA done for these data. It appears there were no significant differences pre-post for any risk categories. Is this true?

Discussion

Line 238- you state in the first sentence that you sought to determine a correlation… You did not perform any correlation analysis, no r values are reported. Please better refine the description of the analysis you performed. Do not use the term” association” as this implies a correlational analysis, and you did a comparative analysis.

The discussion needs to highlight the key aspects of the study results. I cannot find any of that in this discussion. What are the key findings?

Author Response

(The authors gave the same response as above.)

Round 2

Reviewer 1 Report

Comments and Suggestions for Authors

I suggest a prospectively collected data not just on 6MWD and baseline data as included in this study, but also assessment of frailty and markers of nutrition as well as the holy grail which is the VO2 max. for this assertion to be more convincing.

Author Response

I suggest a prospectively collected data not just on 6MWD and baseline data as included in this study, but also assessment of frailty and markers of nutrition as well as the holy grail which is the VO2 max. for this assertion to be more convincing.

Response: Revisions have been made as suggested

Reviewer 2 Report

Comments and Suggestions for Authors

Thank you for the responses.

I have a minor, yet potentially significant, comment:

1. The title indicates that the study focused exclusively on “males.” However, this is not accurate. The abstract does not clarify the issue of missing data related to women. If a reader reads only the title and abstract in their current form, it would be impossible to understand what actually occurred in this study. I suggest that the authors consider a new title, perhaps one that does not mention “males,” and that they clarify in the abstract that missing data affected this research.

Author Response

Thank you for the responses.

I have a minor, yet potentially significant, comment:

1.The title indicates that the study focused exclusively on “males.” However, this is not accurate. The abstract does not clarify the issue of missing data related to women. If a reader reads only the title and abstract in their current form, it would be impossible to understand what actually occurred in this study. I suggest that the authors consider a new title, perhaps one that does not mention “males,” and that they clarify in the abstract that missing data affected this research.

Response: Revisions have been made as suggested

Reviewer 3 Report

Comments and Suggestions for Authors

Revision review comments:

Title now reads: “Differences in 6- minute walk distance across heart disease recurrent risk levels in male cardiac rehab patients”

-          However, results are reporting data that include females. Maybe take out identifying a specific gender in the title.

Abstract line 10- This study seeks…. (language should be in past tense).

-Can you identify total sample size used within the abstract also?

Line 74- Make sure that the tense you use in the paper is past tense. … the objective of this analysis is was to determine…

You have two  table 3s. The first table 3 is hard to visualize. I really think your key data should be in figure form. The primary interest is to examine the pre-post 6mW distances across the 3 risk categories. Since the women were all in the low-risk category, does breaking data out by gender matter as much? Instead, a figure that is perhaps a bar graph showing pre-post data across each of the three risk categories for overall show the effect related to your title.

Table 3 number two- I see a significant p value for main effects. Therefore, the next step would be a post-hoc analysis. Did you do this? Is high> moderate? High> low? moderate> Low?

Discussion- I would make a decision on whether to include female data or not include. Your title suggests that you only included men, but you did include women, but not in the statistical analysis. A decision is needed. Due to small sample size, my suggestion is to delete all of the female data. You can always make mention of the reason behind only assessing men in this study (due to lack of adequate sample size and lack of distribution across all risk categories.

Author Response

I truly appreciate your careful attention to details and valuable feedback to enhance the reasoning and clarity of this work. Thank you so much for taking the time to review my grammar and helping to improve the clarity and quality of this work. Please see attached responses to your recommendations, thanks again.
